# Differences in Growth Performance and Meat Quality between Male and Female Juvenile Nile Tilapia (*Oreochromis niloticus*) during Separate Rearing

**DOI:** 10.3390/ani14202954

**Published:** 2024-10-14

**Authors:** Mohamad Sayouh, Masoud Ali, Yan Li, Yi-Fan Tao, Si-Qi Lu, Jun Qiang

**Affiliations:** 1Freshwater Fisheries Research Center, Chinese Academy of Fishery Sciences, Wuxi Fisheries College, Nanjing Agricultural University, Wuxi 214081, China; mohamadsayouh3@gmail.com (M.S.); liyan@ffrc.cn (Y.L.); taoyf1992@gmail.com (Y.-F.T.); lusiqi@ffrc.cn (S.-Q.L.); 2Research Center of the General Authority for Fish Resources and Aquatic Life, Tartous 96343, Syria; 3Zanzibar Fisheries and Marine Resources Research Institute, Zanzibar 2789, Tanzania; 4Key Laboratory of Freshwater Fisheries and Germplasm Resources Utilization, Ministry of Agriculture and Rural Affairs, Wuxi 214081, China

**Keywords:** Nile tilapia, growth performance, meat quality, RAS system

## Abstract

**Simple Summary:**

This study compared the growth and meat quality of juvenile male and female Nile tilapia. Males grew bigger and heavier, while females excelled in feed conversion. Males had more crude fat and ash, but moisture and protein levels were unchanged. Amino acid profiles were similar overall, though females had an advantage in non-essential amino acids.

**Abstract:**

This study compared the growth and flesh quality of juvenile male and female Nile tilapia grown in separate RAS tanks. The Genetic Sex Determination method yielded 40 males and 40 females. Males grew longer and heavier than females, and the results showed significant variation between the sexes in terms of weight gain rate (WGR), specific growth rate (SGR), and final body length and weight. In terms of the feed conversion ratio (FCR), Gonad Somatic Index (GSI), and Viscera Somatic Index (VSI), females presented better results than males. Male and female Hepatosomatic Index (HSI) values did not differ significantly—no difference in blood serum values. The meat’s moisture and crude protein composition did not alter significantly from male to female; however, the crude fat and ash levels did differ significantly. Male and female animals were given the same seventeen distinct types of amino acids, and there was no distinct variation in the profiles of total amino acids (TAAs) and essential amino acids (EAAs) between the sexes. However, in the non-essential amino acid (NEAA) category, there were marginally significant differences, wherein females performed better than males. Males and females differed considerably in crude fat and ash levels but not in the moisture content or crude protein composition of the meat. Regarding fatty acids, males outperformed females in terms of total fatty acids (TFAs), polyunsaturated fatty acids (PUFAs), and saturated fatty acids (SFAs). However, no significant difference in the amount of monounsaturated fatty acids (MUFAs) in muscle was found between males and females.

## 1. Introduction

The three main species of farmed freshwater finfish are carp, tilapia, and salmonids. Of these, tilapia is the most frequently cultivated, accounting for 71% of the world’s total production [1]. This significant fish is farmed across 120 countries in both freshwater and brackish environments [2]. The success of tilapia production is influenced by a variety of internal and external factors, including genetics, age, sex, water quality, and stocking density [3], all of which directly affect growth performance and meat quality [4,5]. Notably, tilapia’s advantageous traits, such as high stocking density tolerance and adaptability to low-cost diets, facilitate its widespread farming.

However, despite these benefits, early breeding and the tendency for overpopulation in mixed-sex cultures pose significant challenges. This early maturation often diverts energy and nutrients from growth to reproduction, particularly affecting female tilapia. Consequently, farmers increasingly consider cultivating predominantly male tilapia or employing sex-reversal techniques to mitigate these issues [6].

The impact of rearing practices on tilapia growth and meat quality is pronounced when comparing same-sex and mixed-sex cultures. A study by Dawood [7] highlighted that tilapia reared in same-sex environments exhibit more uniform growth patterns, reducing the occurrence of undersized or unsellable fish. While much research has examined why male tilapia tend to grow larger than females, fewer studies have explored the implications of same-sex rearing on growth performance and meat quality.

Comprehending the life cycle of tilapia is essential since every stage from the egg to the adult has an impact on quality and growth. The quality and temperature of the water have an impact on survival rates, with early stages being especially vulnerable. Optimal conditions result in healthier fish and increased productivity. The development of muscular bodies and quality meat in juvenile tilapia is mostly dependent on their nutrition and the density of their stockings [4,8,9]. A healthy diet promotes growth and improves the flavor and texture of meat. Male and female differences become significant when an individual reaches adulthood. While females tend to concentrate more on reproduction, males usually grow quicker and produce more meat. The quality and efficiency of meat production are impacted by this sexual dimorphism [10,11,12,13,14].

The quality of tilapia meat encompasses several physical and chemical attributes, including color, moisture, fat cover, protein content, shelf life, and spoilage potential [15]. Juvenile tilapia, known for their white, firm meat and delicate flavor, are appealing to consumers due to their nutritional value, which includes essential moisture, protein, and fatty acids [15]. However, it is worth noting that farmed fish sometimes exhibit lower-quality meat compared to their wild counterparts. Intrinsic factors such as race, age, and gender, along with environmental conditions like water quality, significantly influence meat quality [16]. Studies have shown that male tilapia often have higher body weights and fillet yields, making them advantageous in aquaculture [11]. Furthermore, maturation leads to distinct differences in fatty acid composition between the sexes. Larger, mature tilapia typically have softer meat with higher fat content compared to their leaner, younger counterparts [17,18,19].

The rearing system also affects meat quality; for instance, recirculating aquaculture systems (RASs) provide more controlled conditions than traditional pond or cage culture, leading to more consistent meat quality [9,20]. Conversely, outdoor and indoor pond cultures can introduce variability based on environmental factors and management practices.

By determining which sex grows faster or produces higher-quality meat, farmers can optimize resource distribution, such as feed and space, thereby enhancing overall productivity. Insights into the nutritional makeup and quality of meat from each sex can inform breeding strategies aimed at improving desirable traits, ensuring a superior product for consumers. These comparisons are essential for advancing tilapia farming practices, ultimately increasing both quality and output. Given the gaps in existing research, this study aims to investigate the differences in growth, blood serum profiles, meat quality, and amino and fatty acid compositions between male and female Nile tilapia reared separately.

## 2. Materials and Methods

### 2.1. Ethics Statement

These experiments were approved by the Bioethical Committee of the Freshwater Fisheries Research Centre (FFRC), Chinese Academy of Fishery Sciences (2013,863 BCE). The regulations for the handling and use of laboratory animals were followed at all times.

### 2.2. Experimental Fish

The juvenile Nile tilapia that was used in the experiment were raised in the Wuxi Fisheries College of Nanjing Agricultural University. The juveniles were obtained and randomly selected from pure brood stock genotypes of males (XY) and females (XX). The average weight and length (±SD) of the juveniles were 22.50 ± 0.31 g and 8.3 ± 0.12 cm, respectively. Before starting the sex determination procedure, the fish were acclimatized in an indoor tank (0.8 m^3^), monitored, and fed for satiation twice a day (9 a.m. and 4 p.m.) for 10 days to ensure that the fish were safe and in good health. Then, twenty experimental fish were taken randomly and used to conduct a sex determination experiment.

### 2.3. Sex Determination

The fish were placed in small tanks of 20 L each (two fish in the tank), connected to a closed filtered water cycle where the water temperature (26 ± 1 °C) and dissolved oxygen (≥6.5 mg/L) were controlled. The sex determination experiment was repeated several times in short order to accurately obtain the required fish (40 males, 40 females). The sex of both males and females was determined by extracting DNA from a small piece of caudal fin (0.5 × 0.5 cm) using the TIANamp Genomic DNA Kit (TIANGEN BIOTECH, BEIJING Co., Ltd., Beijing, China) following the manufacturer’s recommended protocol. Before starting the PCR test, a device (NANODROP LITE—Spectrophotometer—Thermo Scientific Co.—Wilmington, DE, USA) was used to gain the proper concentration of DNA purity optical density (OD) 1.9–2.1. The PCR was conducted with a total reaction volume of 25 μL containing 1 μL of DNA, 1 μL for each primer F and R (Forward and Reverse primers), 12.5 μL of Master Mix II, and 9.5 μL of RNase-free water. Afterwards, the tubes were placed in a microcentrifuge for 10 s and put into a PCR device (BIO-RAD T100TMThermal cycler—Singapore) as follows: initial denaturation (95 °C for 3 min), 35 cycles of denaturation (95 °C for 15 s), annealing (60 °C for 20 s), extension (72 °C for 5 min), and final extension (12 °C for 5 min). After that, agarose gel electrophoresis was conducted, whereby 1.6 g of agarose powder was weighed and placed in a small glass beaker, and then 80 mL (TBE solution) was added and heated in the microwave for 2 m, after which it was cooled in a water bath for 10 s. Then, 0.8 µL of super stain was added into the beaker and mixed with the solution, which was subsequently placed in the template that contained the comb inside it at one end, and finally, left for about 15–20 min until it cooled down and became a gelatinous solid so that the teeth of the comb left the gaps (wells) in which the DNA samples were placed. A suitable micropipette was used to put 0.4 µL DNA samples in the wells. The agarose gel was placed in an electrophoresis device (BIO-RAD, FW Version:1.29-Sn.017797, Singapore), and at one end of the plate, the gel contained gaps called wells, in which the samples were placed. The DNA marker (DL 2000) was placed in the first gap (wells) of the agarose gel to measure the size of the DNA fragments in the other wells. The power supply was set to 100 V for an hour and then the movement of dyed DNA molecules through the agarose gel could be seen. After electrophoresis was completed, a device ultraviolet UV Transilluminator (WD-9413C Gel Imaging Analyzer—Beijing, China) was used to examine the agarose gel using a wavelength of 300 nm to analyze the results.

### 2.4. Experimental Design

After completing the sex determination procedure and separating the males and females, the fish were transferred to the greenhouse, where the experiment was conducted. The initial weight and length of the fish were measured individually and the fish were placed in eight fiber tanks, each with a capacity of 200 L, connected to a recirculating aquaculture system (Zhongkehai Recirculation Aquaculture Systems Co., Ltd., Qingdao, China). The tanks were divided into two groups: four barrels were allocated to the females, and four barrels to the males. The stocking density of each tank was 10 fish. The experiment lasted 85 days, and all groups were subjected to the same experimental conditions in terms of water temperature (27 ± 1 °C), dissolved oxygen (>7 mg/L), pH (7.6 ± 0.2), feed quality, and daily feeding frequency (see Section 2.5).

### 2.5. Fish Feed

In this study, the fish were fed on floating feed; the feed used was manufactured by the company Liu Xinixin Tiansi Aquatic Feed Co., Ltd. (Jiaxing, China) according to the following specifications: the percentage of crude protein was ≥28, the percentage of fat was ≥4, the ash was ≤15, the humidity was ≤10%, phosphorous (1–3), lysine ≥ 1.5, and the diameters were 1.5, 2 and 3 mm and proportional to the stages of growth during 85 days of the experimental period. The fish were fed by hand at two time periods: 8:00–9:00 a.m. and 16:00–17:00 p.m. The feeding ratio is 5% of the calculated body weight based on FCR.

### 2.6. Sample Collections and Analyses of Growth Parameters

After 85 days of fish rearing, the fish were starved for 24 h to clean the intestines before sampling. Then, the fish in each tank were individually weighed by electronic scale to determine the final body weight (FBW), and the final body length (FBL) of both males and females was measured with a measuring plate to the nearest (0.1 cm). The growth performance parameters (SR, WGR, SGR, FCR, GSI, HSI, and VSI) were calculated by using the following formulas:$$Survival\;rate\;(SR)=\frac{Final\;number\;of\;fish}{Initial\;number\;of\;fish}\times 100$$
$$Weight\;gain\;rate\;(WGR)=\left[\frac{Final\;weight\;(g)-initial\;weight\;(g)}{Final\;Weight\;(g)}\times 100\right]$$
$$Specific\;growth\;rate\;\left(SGR\right)=\left[\frac{\ln(final\;weight)-\ln\left(initial\;weight\right)}{Time\;(days)}\times 100\right]$$
$$Feed\;conversion\;ratio\;\left(FCR\right)=\frac{Fish\;feed\;intake}{\left(Final\;body\;weight-Initial\;body\;weight\right)}$$
$$Viscesromatic\;Index\;(VSI)=\frac{Visceral\;mass\;(g)}{Fish\;mass\;(g)}\times 100$$
$$Hepatosomatic\;Index\;\left(HSI\right)=\frac{Liver\;mass\;\left(g\right)}{Fish\;mass\;\left(g\right)}\times 100$$
$$Gonad\;Somatic\;Index\;(GSI)=\frac{Gonad\;mass\;(g)}{Fish\;mass\;(g)}\times 100$$

### 2.7. Sample Collections for Blood Serum Analyses

Before sampling took place, all fish were given 100 mg/L MS-222 (Argent Chemical Laboratories, Redmond, WA, USA) as an anesthetic. The five blood samples (*n* = 5) were from five randomly chosen fish in each tank, and the caudal veins yielded about 1.5 mL of blood. The blood samples were put in centrifuge microtubes and chilled to 4 °C for three hours before serum extraction. The blood samples were then centrifuged at 5000 rpm for 15 min at 4 °C. The sera were drawn out, placed in fresh micro tubes, and sent to Shanghai Langdun Biotechnology Co., Ltd. (Shanghai, China) for analysis. Glucose was determined using the o-toluidine method; total cholesterol was determined using a kit (enzymatic method); and enzymes such as alanine aminotransferase (ALT), aspartate aminotransferase (AST), glucose (GU), and lipids such as total cholesterol (TC) and triglycerides (TGs) were measured using a Mindray BS-400 automatic biochemical analyzer (Shenzhen, China). All test kits came complete with all of the instruction protocols from (Shenzhen Mindray Bio-Medical Electronics Co., Ltd., Shenzhen, China).

### 2.8. Meat Sample Collection

Three fish were chosen at random from each tank, one male and one female (*n* = 6). Using a sharp scalpel, meat samples were extracted from one side of the body starting from the vicinity of the head and ending at the tail. The samples of flesh weighed about 50 g and were sliced free of scales, skin, and spines. Male and female samples were separated using laboratory bags, and the samples were then stored in the freezer (−80 degrees) until usage.

### 2.9. Analysis of Meat Proximate Composition

#### 2.9.1. Moisture Content

The typical standard drying temperature used for moisture content was 105 °C (221 °F) and the sample was dried at this temperature for 16–24 h. According to the AOAC [21], the moisture content was estimated using the oven drying method by weighing the sample both before and after the water was removed and using the following formula:$$Moisture\;content=\frac{Weight\;of\;wet\;sample\;\left(g\right)-Weight\;of\;dried\;sample\;(g)}{Weight\;of\;wet\;sample\;(g)}\times 100$$

#### 2.9.2. Crude Protein

Following the AOAC technique [21], the amount of crude protein in each fish’s meat sample was determined using the Kjeldahl method.

#### 2.9.3. Crude Fat

Crude fat was quantified using the Soxhlet method, a semi-continuous solvent extraction technique, in line with the AOAC procedure [22]. The crude fat content of the initial sample was calculated as follows:$$Fat\;content=\frac{Weight\;of\;the\;fat\;(g)}{Weight\;of\;sample\;(g)}\times 100$$

#### 2.9.4. Ash Content

The AOAC method [23] was applied to ascertain the ash content. Below was how the ash content was determined:$$Ash=\frac{\left(WAA-TWC\right)}{Original\;sample\;weight}\times 100$$
whereby WAA is the weight after ashing and TWC is the tare weight of the crucible.

#### 2.9.5. Analysis of Amino and Fatty Acids

##### Amino Acid Analysis

High-performance liquid chromatography (HPLC; Shimadzu Corp., Beijing, China) was used to analyze amino acid levels. After being spiked with a known amount of norleucine as an internal standard, 2 mg of the fish samples was hydrolyzed with 4 N methanesulfonic acid at 110 °C for 22 h. After filtering, the hydrolysate’s pH was adjusted to 2.2 and it was then maintained at 4 °C. The chromatographic separation and analysis of the amino acids were performed, according to [24], using an HPLC system with an ion exchange resin column.

##### Fatty Acid Analysis

All lipids were isolated from meat samples and homogenized in chloroform/methanol (2:1, *v*/*v*) to evaluate the fatty acids. Fatty acid methyl esters (FAMEs), which were extracted in heptanes, analyzed using a gas chromatograph (7890A, Agilent, Beijing, China), and outfitted with an auto-sampler and a hydrogen flame ionization detector, were created by methylating in 1% sulfuric acid at a temperature of 70 °C for three hours. A 30 m × 0.25 mm × 0.25 μm capillary column (VF-23 ms, Varian, Salt Lake City, UT, USA) was installed in the gas chromatograph. N2 served as the carrier gas, and air and H2 served as the combustion-supporting gases. The injector and detector both had temperatures of 250 °C. The column temperature was initially maintained at 120 °C for three minutes before being raised by 10 °C/min to 190 °C. Once the temperature reached 220 °C, it was sustained for 15 min at a rate of 2 °C/min. Using the Sunny Hengping FA-1004 device (Shanghai Sunny Hengping Scientific Instrument Co., Ltd., Shanghai, China), fatty acids were individually recognized and measured by comparison with commercial standards (Sigma, Salt Lake City, UT, USA) [25].

### 2.10. Correlation Analyses

Correlation analysis was performed to find out which indices (blood serum indices, meat quality) show the most relation with certain growth performances. The correlation was analyzed using the Origin pro-2021-correlation plot.

### 2.11. Data Analysis

Statistical data analysis was performed using the statistical package for the social sciences (IBM SPSS Statistics 19, New York, NY, USA) to conduct paired sample *t*-test on the data, where *p* < 0.05 indicated a significant difference and the data are presented as mean ± standard deviation (mean ± SD).

## 3. Results

### 3.1. Genetic Sex Determination

After the process of electrophoresis and examining the agarose gel by a UV Transilluminator to obtain the results, both male and female Nile tilapia are identified by observing the difference in the number of bands, as shown in Figure 1, where females have only X chromosomes, and therefore one DNA band, while males have both X and Y chromosomes, and thus amplification by PCR produces two DNA bands. It is noticed that the percentage of males in each repetition is higher than the percentage of females; the columns (4, 6, 8, 11, 12, 13, 14) show that there is one band that indicates the female gene (XX), while the other columns (2, 3, 5, 7, 9, 10, 15, 16) show two bands, indicating the male gene (XY) at the same bp (1500).

### 3.2. Growth Performance Parameters

The growth performance parameters of juvenile male and female Nile tilapia reared in separate tanks at 85 days are summarized in Table 1. There is no significant difference in HSI between males and females (*p* > 0.05), while there are significant differences in FBL, FBW, WGR, SGR, FCR, GSI, and VSI between males and females (*p* < 0.05).

### 3.3. Blood Serum (ALT, AST, GLU, TC, TG)

The results in Table 2 below show that there was no significant difference (*p* > 0.05) in the following serum parameters between males and females during the stage of sexual maturity: alanine aminotransferase (ALT), aspartate aminotransferase (AST), glucose (GLU), total cholesterol (TC), and triglycerides (TGs).

### 3.4. Approximate Composition of Meat

The results of the approximate composition of meat in Table 3 below indicate that there is no significant difference (*p* > 0.05) in the moisture and crude protein between male and female Nile tilapia. However, there are significant differences (*p* < 0.05) in crude fat and ash content between male and female Nile tilapia, as the results in Table 3 for fat are 1.30 ± 0.17%, 1 ± 0.10%, and ash content are 1.13 ± 0.05%, 0.76 ± 0.06%, for male and female, respectively, where the fat and ash contents in males are greater than females.

### 3.5. Profile of Amino Acids

In this study, 17 different types of amino acids were obtained in both male and female Nile tilapia. The results are shown in Table 4, wherein the total content of each amino acid is presented as (g/100 g protein) and eight of the nine essential amino acids for the human body have been identified. The results showed that there was no significant difference (*p* > 0.05) between males and females in the profile of essential amino acids (EAAs), while there was a low significant difference (*p* < 0.05) between males and females in muscle content of non-essential amino acids (NEAAs). The results of the total amino acid (TAA) profile were 15 ± 0.152% and 15.866 ± 0.120% for males and females, respectively, indicating that the TAA in females is slightly higher than in males, but the statistical study showed that there is no significant difference (*p* > 0.05) and no effect of sex on TAA profile.

### 3.6. Profile of Fatty Acids

The results of fatty acids are classified in Table 5 into saturated fatty acids (SFAs), monounsaturated fatty acids (MUFAs), polyunsaturated fatty acids (PUFAs), and total fatty acids (TFAs), where it can be seen that there is a significant difference (*p* < 0.05) in TFAs, SFAs, and PUFAs between males and females, where the results are 1.05 ± 0.045%, 0.628 ± 0.015% for TFAs, 0.381 ± 0.0143%, 0.224 ± 0.0026% for SFAs, and 0.337 ± 0.012%, 0.237 ± 0.007% for PUFAs for males and females, respectively, as males outperform the females. And the results showed that there was no significant difference (*p* > 0.05) between males and females in the content of MUFAs, where the results were 0.327 ± 0.0307%, and 0.167 ± 0.0114% for males and females, respectively. This indicates that MUFAs are slightly higher in males than in females, but the statistical study indicated that there is no effect of sex on the muscle content of MUFAs.

Also, there is no significant difference (*p* > 0.05) between males and females in the muscle content of PUFA n-3 and n-6.

### 3.7. Correlation Analyses

For males, the results of the correlation among growth parameters, blood serum indicators, and meat quality are shown in Figure 2. There was a positive correlation at different levels with no significance *p* > 0.05 between FBL, FBW and protein, ash, fat, ALT, AST, and fatty and amino acids, as there was an inverse correlation at different levels with no significance *p* > 0.05 between WGR, SGR, GSI, FCR, HSI, VSI and ALT, AST, protein, ash, fatty and amino acids.

For females, the results in Figure 3 show that there was a direct correlation at different levels between FBL, FBW and TC, TG, moisture, protein, EAAs, TAAs, SFAs, MUFAs, and TFAs, while there was an inverse correlation between WGR, SGR, FCR, and GSI and moisture, protein, EAAs, TAAs, SFAs.

## 4. Discussion

### 4.1. The Growth Performance

In this study, males outperformed females in WGR and SGR; this indicates the efficiency of males in converting the bulk of their energy from digestive processes for physical growth [14,26,27] mentioned that the majority of the metabolic energy in Nile tilapia males resulting from the consumption of the provided feed is directed towards growth and masculinization, whereas the majority of the metabolic energy in females is typically directed towards reproduction and egg formation, which results in slow growth. The FCR of male tilapia was lower than females (Table 1) [28]. This indicates the efficiency of males in converting feed and benefiting from it in weight gain compared with females [29]. Moreover, Osibona [30] reported that, despite eating the same number of feeds, males grow bigger and more rapidly than females due to their superior ability to convert food into energy. This result shows a significant correlation between growth and FCR, whereby faster-growing fish had a better lower FCR.

In this study, the GSI of females was higher than that of males, as the female gonads were fully mature and full of eggs. Fleming [31] notes that female fish may invest 20–25% of their body weight in gonads before reproduction and require a great deal of energy to develop eggs, but male fish may invest only 3–9%. When the male gonads reach the ripe (running) stage, the fish’s ripe gonads mark a critical moment when energy is shifted towards successful reproduction [32]. Therefore, the mature state of male gonads affects the quality of meat and causes changes in fat content, texture, flavor, appearance, and shelf life, all of which can affect consumer acceptance and marketability. In addition, there was a positive correlation between HSI and GSI during the development of the gonads [33], where raised HSI may help maintain the transport of nutrients to the developing ovary, increase GSI, and promote follicle [34]. However, there was no significant difference in HSI between males and females, as the slightly elevated HSI in females may be due to changes in energy intake and homeostasis during ovarian development, ovulation, and reproductive processes.

### 4.2. Blood Serum Indices

There was no significant difference between both males and females in blood serum parameters, as it reflects the healthy state of the fish. Perhaps the reason is that all the fish were subjected to the same parameters of the experiment and were not exposed to any stress during the experimental period. As mentioned by Heiba [35], changes in blood serum parameters are caused by fluctuations in water quality parameters, the presence of contaminants in the water, various types and proportions of feeds, and the composition of the protein in the feed. This study is consistent with the study conducted by Fagbuaro [36] which found that in Nile tilapia, sex did not affect blood serum components (ALT, AST, TC, TG); rather, the difference was in the concentration of glucose (GLU), whereby in males, it was higher than in females.

The values of TC and TG in the current study do not agree with the values of Fagbuaro [36], which were considered very low. No significant difference in total cholesterol was found between males and females. Maita [6] indicated that fish genotype significantly influences total cholesterol, which is associated with disease resistance in fish, whereas Fagbuaro [36] indicated that the difference in total cholesterol concentrations in Nile tilapia resulted from different activities, environmental conditions, and habitats, and as in our current study, the obtained values did not show any significance between the two sexes.

No significant difference in HSI was seen between males and females (Table 1), and one factor that might account for this is the presence of the same levels of the hormones ALT and AST, which are frequently used as markers of hepatocyte injury or as indicators of vertebrate liver activities. This indicates that both the male and female livers are healthy and function normally as ALT and AST are generally considered biomarkers of liver and kidney health [37]. There is a difference in the correlation between ALT, AST, and HSI in males and females, as it is inverse in males, as shown in Figure 2, and positive in females, as shown in Figure 3, at the significance level.

Blood glucose is considered one of the most important sources of energy for the cells of the body, as the level of glucose in the blood is maintained through the digestion of dietary carbohydrates [38]. There was an inverse correlation between ALT, AST and GLU, TC in males, as shown in Figure 2, while the correlation was positive in females, as shown in Figure 3, at the significance level. However, the absence of a variation in glucose content between the sexes may be viewed as a source of energy consumed by both sexes in an equal distribution during the gonads’ stages of growth and development, where the oxidation of fatty acids provides the necessary energy for fish metabolism, which may require glucose [39].

### 4.3. Meat Quality (Approximate Composition)

#### 4.3.1. Moisture Content

No significant difference in the moisture content of meat was found in both males and females, as shown in Table 3, and the results were low compared to many previous studies. Also, there was an inverse correlation between moisture and FBL in males, as shown in Figure 2, while a positive correlation was seen in females, as shown in Figure 3, at the significance level. The results of this study do not agree with the results reported by Amer [39] which state that females contain less moisture than males. In contrast, Olopade [40] reported that the moisture in the meat fillets of both male and female Nile tilapia was 81.11% ± 2.76 and 81.67% ± 1.66, respectively, and this moisture content is very high compared to the current study. This low moisture content in the meat plays a role in preserving the characteristics, quality, and taste of the meat. Olagunju [41] stated that if the moisture content of the meat is greater than 80%, it is considered to be high in moisture, which makes fish meat and fish products more susceptible to microbial spoilage and oxidative decomposition of polyunsaturated fatty acids in addition to lowering the quality, safety, and shelf life of these products.

#### 4.3.2. Crude Protein

There is no significant difference in crude protein in the composition of the meat between males and females. While the crude protein content of fish muscles ranges from 15% to 28%, some fish species can have values as low as 15% or as high as 28% [16]. According to the results in Table 3, the protein content was greater than 15% for males and females; thus, the meat of Nile tilapia is a rich and high source of crude protein, as reported by Stancheva [42]. Additionally, in this study, the crude protein was high compared to many previous studies, such as the works of Olopade [40] and Alemu [43], which reported that the crude protein was 13.25%, 14.26% and 14.5%,14.6% for males and females, respectively. The age of the fish from which the meat samples were taken may be one reason for this difference in crude protein in the approximate composition of meat between the current and previous studies; for example, Cornelia [44] described that juvenile fish contain more protein than adult fish. According to Abdel-Tawab [45], muscle proteins tend to increase with an increase in the level of dietary protein used in the manufacture of feed used to feed Nile tilapia, so the high crude protein in this study may also be related to the type of feed that was used, in which the percentage of protein in feed was greater than 28%. In this study, the correlation between protein and moisture was also inverse in males, as shown in Figure 2, while it was positive in females, as shown in Figure 3, at the significance level.

#### 4.3.3. Crude Fat

In the current study, there is a significant difference between males and females in the content of crude fat, as shown in Table 3. The crude fat is higher in males compared to females, which causes it to be considered as lean fish according to the classification by Ackman [1], who reported that fish can be divided into four categories based on their crude fat content: lean fish (crude fat percentage < 2%), i.e., low-fat fish; the percentage of crude fat between (2 and 4%); medium fat (4% to 8%); and high fat (if the percentage of crude fat > 8%). Perhaps one of the reasons for the decrease in crude fat in this study is that the fish are still in the stage of puberty and sexual maturity, as the fat content is affected by species, sex, geography, and seasons, whereby females use more energy than they derive from the breakdown of fats in the development of gonads and the formation of eggs [46,47]. As Caponio [48] mentioned in the case of gonad development and ovulation, the fat content of fish may change because males and females at these stages consume energy differently, with lipids serving as the main source.

In a previous study, Olopade [40] reported that the crude fat in Nile tilapia was 0.57% + 0.07 and 0.5% + 0.08 for males and females, respectively, which is lower than the results of the current study. Although researchers, including Alemu [43] and Massresha [49], suggested an inverse relationship between moisture content and crude fat in the composition of fish meat, it was observed in this study that a high moisture content was associated with a relatively low crude fat content, and this was consistent with the correlation plot in this study for both males and females, as shown in Figure 2 and Figure 3, respectively. This may have been due to the low moisture content in both males and females in this study.

#### 4.3.4. Ash Content

There is a significant difference in the ash content in the composition of the meat, which is higher in males than females. This indicates that the meat of males contains more minerals than females, since the percentage of ash reflects the amount of minerals in the sample [50]. However, the percentage of ash is related to the breeding environment and the vital processes of the fish body. Also, there is a positive correlation between ash and moisture in males, as shown in Figure 2, while an inverse correlation is seen in females, as shown in Figure 3, at the significance level. Olopade [40] mentioned that the ash content for males and females was 1.24% ± 0.15 and 1.48% ± 0.39, respectively, and the findings of the ash study by Alemu [43] were 1.14% in males and 1.17% in females. The two studies are consistent with the results of the current study for males but differ with the ash content in females.

### 4.4. Profile of Amino Acids

Nile tilapia is regarded as a fish that is rich in essential amino acids Fleming [31]. No significant difference between males and females in the TAAs was detected, as well as in the EAAs because proteins are the basic constituents of cells during the early stages of growth, differentiation, development, and sexual maturity, synthesized by both males and females in proportionate amounts. However, the number of amino acids in tilapia fluctuates based on a variety of factors, such as the habitat, the type and composition of the fish’s diet, the fish’s age, weight, and length [7]. The FAO and WHO reported that high-quality dietary protein not only contains a full range of essential amino acids, but these essential amino acids are also in appropriate proportions [50]. As determined by the WHO/FAO (2007), protein is considered to be of high quality when the ΣEAA/ΣTAA ratio is more than 40%, and the ΣEAA/ΣNEAA ratio is more than 60%. In this study, the ratio of ΣEAA/ΣTAA is 45.35% and 45.22% in males and females, respectively, and higher than 40%; and the ratio of ΣEAA/ΣNEAA is 83.33 and 82.86 in both males and females, respectively, and much higher than 60%. Thus, the protein in the meat of Nile tilapia is of high quality, and the protein quality of males is slightly higher than that of females based on this indicator.

The content of lysine is the highest in the essential amino acids, as shown in Table 4, where the value of lysine is 1.570% ± 0.020 and 1.653% ± 0.008 for both males and females, respectively, while the content of methionine is the lowest of the essential amino acids, with an amount of 0.396%, which is similar in both males and females.

### 4.5. Profile of Fatty Acids

In this study, there is a significant difference in the content of the muscles of males and females in each of the SFAs, PUFAs, and the TFAs that were determined, where the males outperformed the females. The difference in fatty acid composition in Nile tilapia is due to differences in stages of sexual maturity, vital processes, and metabolic processes between the sexes [51]. These results are consistent with some results and differ from other results according to some previous studies. As reported by Abelti [8] and Sun [52], the content of SFAs and MUFAs in female muscles is higher than that of males, and it differs from the results of the current study, but the agreement in the results is that male muscles contain a greater amount of PUFAs and TFAs compared to females. The samples used in this study came from fish that were sexually mature, and it is possible that the females consumed the majority of the fatty acids necessary for the development of the ovaries and the production of eggs. The fatty acids in the female fish’s fat are transmitted to the eggs during the growth and development of the gonads in the bloodstocks, and they play a significant role in determining the quantity and quality of the eggs.

As reported by Abelti [8], palmitic acid as a major metabolite is not affected by diet in fish, and it is a dominant SFA in Nile tilapia and found in high concentrations in the adipose tissue; this is consistent with the results of the current study, but it is slightly higher in males than females.

In the current study, males had slightly higher levels of MUFAs in their muscles than females, but statistically, there was no significant difference between the sexes. This contrasts with the work of Abelti [8] which reported that females had slightly higher levels of MUFAs in their muscles than males, possibly because samples were collected during sexual maturity and before ovulation in this study, and females consumed more MUFAs than males during the process of ovarian development and formation of eggs. In this study, not all PUFAs, especially n-3 and n-6, were identified. Perhaps the reason for the low percentages was caused by the increased consumption by both males and females as PUFAs are of great importance in the growth, development, and sexual maturity of juvenile fish. There was a strong positive correlation between PUFAs and WGR in females, as shown in Figure 3, with the significance level, while the correlation was inverse in males, as shown in Figure 2, with the significance level. There is no effect of sex on muscle content of n-3 and n-6 PUFAs; perhaps the reason for this is due to their similar important role for both males and females during the stages of growth and development of the various organs of the body, in addition to the growth and development of the gonads.

## 5. Conclusions

In this research, male Nile tilapia fish perform better than female fish in terms of growth and flesh quality, especially when it comes to growth metrics and fatty acids. Farming profitability and efficiency could be increased by rearing males alone. Promoting sustainable aquaculture methods, such as early Genetic Sex Determination, enables producers to concentrate on male tilapia, improving output and market value.

## Figures and Tables

**Figure 1 animals-14-02954-f001:**
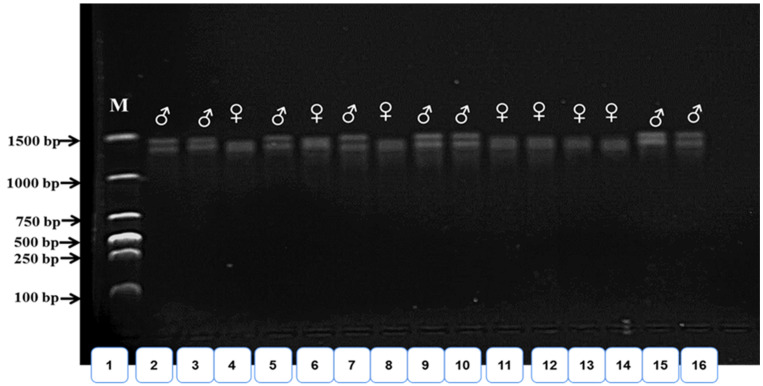
PCR results of GM597 as a sex-specific marker in Nile tilapia (*Oreochromis niloticus*). Female: single band of 127 bp; male: double bands of 127 bp and 137 bp.

**Figure 2 animals-14-02954-f002:**
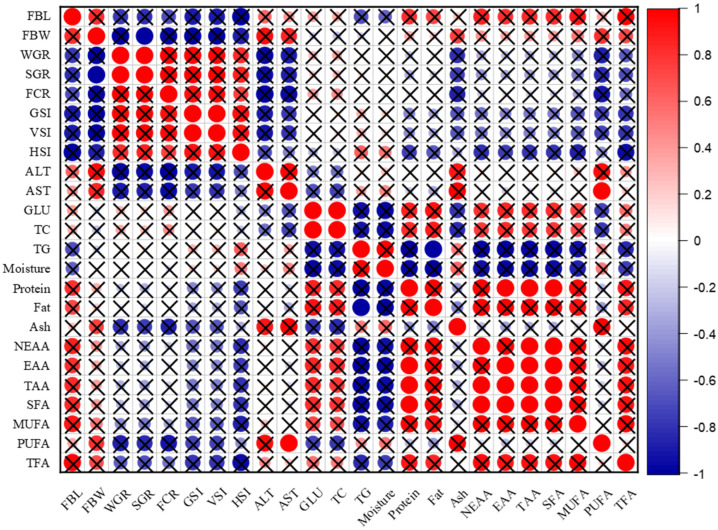
The correlation plot of males between growth parameters, blood serum indicators, and meat quality at a significance level of 0.05. Note: the red color expresses positive correlation, the blue color expresses inverse correlation, and X expresses no significance *p* > 0.05.

**Figure 3 animals-14-02954-f003:**
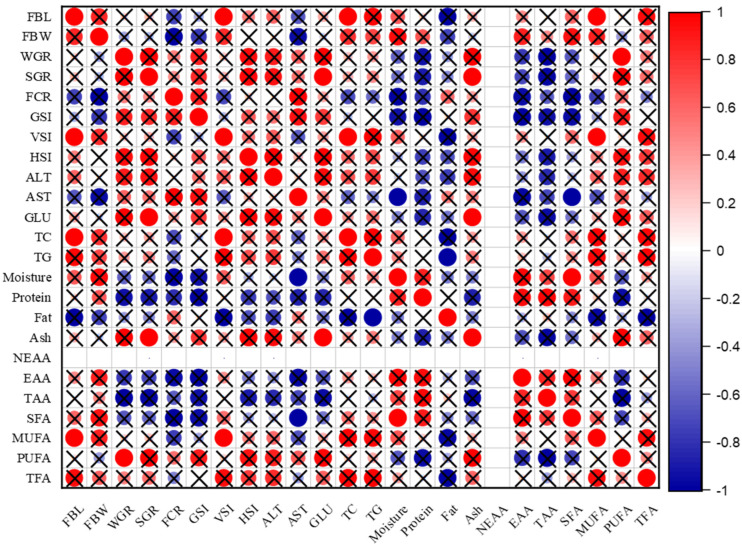
The correlation plot of females between growth parameters, blood serum indicators, and meat quality at a significance level of 0.05. Note: the red color expresses positive correlation, the blue color expresses inverse correlation, and X expresses no significance *p* > 0.05.

**Table 1 animals-14-02954-t001:** Growth performance parameters for male and female Nile tilapia.

Parameters	Male	Female
SR %	100 ± 0.00	100 ± 0.00
FBL (cm)	17.35 ± 0.14 ^b^	16.52 ± 0.01 ^a^
FBW (g)	207.35 ± 0.10 ^b^	173.59 ± 0.01 ^a^
WGR %	88.79 ± 3.95 ^b^	86.7 ± 0.10 ^a^
SGR %	2.58 ± 1.34 ^b^	2.37 ± 0.32 ^a^
FCR	1.56 ± 0.57 ^a^	1.86 ± 1.02 ^b^
GSI %	0.44 ± 0.39 ^a^	3.35 ± 0.35 ^b^
VSI %	8.14 ± 0.06 ^a^	11.55 ± 0.31 ^b^
HSI %	1.79 ± 0.03	2.18 ± 0.05

Note: Data are means ± SD. The values in the different lowercase letters show significant differences between experimental groups within the same row (*p* < 0.05), whereby FBL: final body length; FBW: final body weight; WGR: weight gain rate; SGR: specific growth rate; FCR: feed conversion ratio; GSI: Gonad Somatic Index; VSI: Viscesromatic Index; HIS: Hepatosomatic Index.

**Table 2 animals-14-02954-t002:** Blood serum parameters for male and female Nile tilapia.

Parameters	Male	Female
ALT (nmol/min/mL)	47.652 ± 2.462	46.688 ± 3.682
AST (nmol/min/mL)	40.17 ± 2.989	40.577 ± 1.417
GLU (mg/mL)	0.923 ± 0.021	0.908 ± 0.004
TC (mmol/L)	3.207 ± 0.053	3.136 ± 0.066
TG (mmol/L)	1.379 ± 0.107	1.377 ± 0.115

Note: Data are means ± SD. Whereby AST: aspartate aminotransferase; ALT: alanine aminotransferase; GLU: glucose; TC: total cholesterol; and TG: triglyceride.

**Table 3 animals-14-02954-t003:** The meat quality (approximate composition) for male and female Nile tilapia.

Parameters (g/100 g)	Male	Female
Moisture	76.90 ± 0.40	77 ± 0.05
Protein	19.76 ± 0.21	20.23 ± 0.13
Fat	1.30 ± 0.17 ^b^	1 ± 0.10 ^a^
Ash	1.13 ± 0.05 ^b^	0.76 ± 0.06 ^a^

Note: Data are means ± SD. The values in the different lowercase letters show significant differences between experimental groups within the same row (*p* < 0.05).

**Table 4 animals-14-02954-t004:** Amino acid composition for male and female Nile tilapia.

Amino Acids (g/100 g Protein)	Male	Female
Aspartic acid (Asp)	1.573 ± 0.024	1.686 ± 0.012
Tyrosine (Tyr)	0.466 ± 0.012	0.453 ± 0.008
Serine (Ser)	0.500 ± 0.000	0.510 ± 0.005
Glutamic acid (Glu)	2.133 ± 0.039	2.296 ± 0.008
Arginine (Arg)	0.983 ± 0.008	1.063 ± 0.014
Glycine (Gly)	0.813 ± 0.008	0.893 ± 0.027
Alanine (Ala)	1.026 ± 0.006 ^a^	1.097 ± 0.014 ^b^
Cystine (Cys)	0.100 ± 0.005	0.108 ± 0.007
Proline (Pro)	0.566 ± 0.008	0.616 ± 0.014
ΣNEAA	8.163 ± 0.075 ^a^	8.696 ± 0.075 ^b^
Valine (Val) *	0.889 ± 0.015	0.913 ± 0.008
Methionine (Met) *	0.396 ±0.003	0.396 ± 0.014
Isoleucine (IIe) *	0.783 ± 0.013	0.840 ± 0.005
Leucine (Leu) *	1.370 ± 0.015	1.440 ± 0.010
Threonine (Thr) *	0.720 ± 0.005	0.740 ± 0.0100
Phenylalanine (Phe) *	0.706 ± 0.012	0.750 ± 0.010
Lysine (Lys) *	1.570 ± 0.020	1.653 ± 0.008
Histidine (His) *	0.406 ± 0.006	0.443 ± 0.003
ΣEAA	6.803 ± 0.084	7.176 ± 0.038
Total amino acids	15 ± 0.152	15.866 ± 0.120

Note: Data are means ± SD. The values in the different lowercase letters show significant differences between experimental groups within the same row (*p* < 0.05), where * indicates the essential amino acids.

**Table 5 animals-14-02954-t005:** Fatty males and female Nile tilapia.

Fatty Acids (g/100 g Muscle)	Male	Female
Myristic C14:0	0.0159 ± 0.0006 ^b^	0.009 ± 0.0009 ^a^
Palmitic C16:0	0.259 ± 0.0106 ^b^	0.145 ± 0.002 ^a^
Stearic C18:0	0.107 ± 0.003 ^b^	0.069 ± 0.0009 ^a^
ΣSFAs	0.381 ± 0.0143 ^b^	0.224 ± 0.0026 ^a^
Palmitoleat C16:1	0.0214 ± 0.0014	0.0121 ± 0.0013
Cis-11-Eikosanoic, C20:1	0.0174 ± 0.0021	0.008 ±0.0003
Oleic C18:1n9c	0.289 ± 0.0272	0.147 ± 0.0102
ΣMUFA	0.327 ± 0.0307	0.167 ± 0.0114
Linoleic C18:2n6c	0.190 ± 0.0103	0.117 ±0.009
γ-Linolenic C18:3n6	0.011 ± 0.0009	0.007 ± 0.0002
Linoleic C18:3n3	0.0131 ± 0.0008	0.008 ± 0.0007
Cis-11,14-Eicosedienoic C20:2	0.0126 ± 0.0007 ^b^	0.008 ± 0.0008 ^a^
Cis-8,11,14-Eicosetrienoic C20:3n6	0.023 ± 0.002 ^b^	0.015 ± 0.0006 ^a^
Erucic Acid Methyl Ester, C22:1n9	0.008 ± 0.0009	0.005 ± 0.0004
Arachidonic C20:4n6	0.046 ± 0.0035	0.044 ± 0.0035
Docosahexaenoic C22:6n3	0.035 ± 0.001	0.033 ± 0.003
ΣPUFA	0.337 ± 0.012 ^b^	0.237 ± 0.007 ^a^
Σn-3	0.048 ± 0.002	0.041 ± 0.002
Σn-6	0.079 ± 0.001	0.067 ± 0.004
TFA	1.05 ± 0.045 ^b^	0.628 ± 0.015 ^a^

Note: Data are means ± SD. The values in the different lowercase letters show significant differences between experimental groups within the same row (*p* < 0.05).

## Data Availability

The original data of this paper are available upon request.

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
