# Peer review of "Differences in Growth Performance and Meat Quality between Male and Female Juvenile Nile Tilapia (*Oreochromis niloticus*) during Separate Rearing"

_animals, 2024, doi:10.3390/ani14202954_

Round 1

Reviewer 1 Report

Comments and Suggestions for Authors

The study shows that male tilapia grow more quickly than female fish, gaining more length and weight. Furthermore, the article presents data on the sex differences in the nutritional composition of fish. The results of this study can be potentially useful for aquaculture practices.

Please address the issues listed below.

L17 Please correct “feed convention ratio” to “feed conversion ratio”.

L18 Please correct “men” to “male”.

LL23-29 These sentences repeat the information given above.

L30 “there were marginally significant differences” – please rephrase, the meaning is not clear

L58 remove comma

L64 Unclear sentence. Do you mean that freshwater fish has less nutrients than Nile tilapia?

L72 Please clarify the abbreviation.

L86 “Fish with well-developed genetics”. Please rephrase, unclear sentence.

L151-158. Please type the formulas in the same manner (mind the brackets and commas).

L160 Please specify the anesthesia method.

L162 Mind the reference.

L166 Please specify the other parameters measured.

L219 Mind the reference.

L236 Is it worth mentioning? It seems that 8 to 7 is not a significant excess.

L243 Please check grammar.

L262 “Total sugar” – seems to be a mistake.

L322-333. This text seems to belong to the M&M section. Please explain the rationale for putting it in the Discussion.

While there are some valuable pieces of data in this paper, it is difficult to read due to grammatical mistakes. Therefore, I strongly recommend performing a grammar check throughout the text.

Comments on the Quality of English Language

Please ask any competent person to edit the text or at least run it through a grammar checker, e.g., Grammarly or QuillBot.

Author Response

Dear editors and reviewers,

Thank you for your letter and comments on our manuscript titled " Differences in growth performance and meat quality between male and female juvenile Nile tilapia (Oreochromis niloticus) during separate rearing". These comments helped us improve our manuscript and provided important guidance for future research.

We have addressed the editor’s and reviewers’ comments to the best of our abilities. We hope our manuscript now meets your requirements for publication.

We marked the revised portions in red in our manuscript. All comments and our specific responses are detailed below.

Reviewer 1#

Comment 1: L17 Please correct “feed convention ratio” to “feed conversion ratio”.

Response 1: Already corrected.

Comment 2: L18 Please correct “men” to “male”.

Response 2: Already corrected

Comment 3:

LL23-29 These sentences repeat the information given above.

Response 3: Already deleted and corrected

Comment 4:

L58 remove comma, L30 “there were marginally significant differences” – please rephrase, the meaning is not clear

Response 4: L58 Already deleted and corrected, and L30 was corrected as “However, no significant difference in the amount of monounsaturated fatty acids (MUFA) in muscle between males and females”.

Comment 5:

L64 Unclear sentence. Do you mean that freshwater fish have fewer nutrients than Nile tilapia?

Response 5: L5 meant that “In comparison to smaller fish, which are leaner and tougher, larger tilapia often have softer meat and a higher fat content. In comparison to younger fish, older tilapia tend to store more fat, which results in a higher fat content and a unique texture. Conversely, because the culture method such as the pond culture—has less controlled circumstances, the meat composition may vary.

Comment 6:

L72 Please clarify the abbreviation.

Response 6: Already corrected, DPH (Days Post Hatching)

Comment 7:

L86 “Fish with well-developed genetics”. Please rephrase, the unclear sentence.

Response 7: L81 “The juveniles were obtained from the brood stock with pure genetic”.

Comment 8:

L151-158. Please type the formulas in the same manner (mind the brackets and commas).

Response 8: Already corrected.

Comment 9:

L160 Please specify the anesthesia method.

Response 9: L155 Before sampling took place, all fish were given 100 mg/L MS-222 (Argent Chemical Laboratories, Redmond, WA, USA) as an anesthetic

Comment 10:

L162 Mind the reference.

Response 10: Already corrected

Comment 11:

L166 Please specify the other parameters measured.

Response 11: L 167- enzymes such as alanine aminotransferase (ALT), aspartate aminotransferase (AST), and lipids such as total cholesterol and triglycerides were measured using a Mindray BS-400 automatic biochemical analyzer (Shenzhen, China).

Comment 12:

L219 Mind the reference.

Response 12: Already been corrected, and the reference of L219 is [24].

Comment 13:

L236 Is it worth mentioning? It seems that 8 to 7 is not a significant excess.

Response 13: It is different as is seen in Figure 1, the 8-column indicated one band that indicates the female gene (XX), while the 9- columns showed the two bands indicating the male gene (XY) at the same bp (1500).

Comment 14:

L243 Please check grammar.

Response 14: Already corrected as PCR results of GM597 as a sex-specific marker in Nile tilapia (Oreochromis niloticus). Female: single band of 127 bp; Male: double bands of 127 bp and 137 bp.

Comment 15:

L262 “Total sugar” – seems to be a mistake.

Response 15: Already corrected.

Comment 16:

L322-333. This text seems to belong to the M&M section. Please explain the rationale for putting it in the Discussion.

Response 16: We decided to remove this section from the discussion because the Genetic Sex determination was conducted to get the sexes of experimental fish only and not to discover or to observe something new.

Comment 17:

While there are some valuable pieces of data in this paper, it is difficult to read due to grammatical mistakes. Therefore, I strongly recommend performing a grammar check throughout the text.

Response 17: Already corrected. Finally, thank you to the reviewers for his/her valuable comments, which will be of great help to our next research work.

Reviewer 2 Report

Comments and Suggestions for Authors

The authors comprehensively investigated many parameters related to growth and clarified the differences between males and females. Although the idea is not very novel, there are no particular problems with the method. However, as a scientific paper, there are some issues that need to be addressed, so I would like the authors to consider revising the manuscript.

Overview

1. It is already known that there are differences in growth between males and females in Nile tilapia. I would like authors to emphasize the significance of comparing the differences between males and females in this paper in the Introduction and the Discussion.

2. The discussion is done for each item analyzed, but it is mostly a repetition of results and comparisons with previous studies. I would like authors to clarify what are new findings of this study and want to show from them. To do this, please consider rewriting the discussion significantly.

3. The females used in this analysis have been fully matured, but the condition of the males is not described. It is natural that there would be differences in metabolism between males and females at advanced maturity. Please briefly explain the life cycle of Nile tilapia in the introduction and explain why fish at this developmental stage were selected for the experiment. Also, please state the author's outlook for how he plans to utilize the present results in the future.

4. There are many inappropriate capital letters, so please correct them.

Details

Line 18: men → male?

Lines 17-20 and 23-26: The content is duplicated, so please remove one of them.

Line 64: I think Nile tilapia is also a freshwater fish. Please specify the freshwater fish authors compared.

Lines 76-79: I think it would be better to move this part to the Materials and Methods.

Line 144: Please indicate how often weight measurements were taken to determine feeding amounts.

Lines 152-158: Please unify the format of the equation (with or without parentheses, period or comma, position of %, etc.).

Line 163: If possible, please convert "rpm" to "G".

Line 171: Please make sure that the number is correct.

Lines 179-180: Please indicate the drying temperature and time.

Lines 182, 190: Are the units %?

Table 1, 2: Please add an explanation of abbreviations to the tables as well.

Lines 317-333: This paragraph is a general knowledge and can be removed. If you do write it, please simplify and move to the Materials and Methods.

Lines 356-359: Please also describe the condition of the gonads of male. I think that body composition is greatly affected by the state of maturity.

Author Response

Dear editors and reviewers,

Thank you for your letter and comments on our manuscript titled " Differences in growth performance and meat quality between male and female juvenile Nile tilapia (Oreochromis niloticus) during separate rearing". These comments helped us improve our manuscript and provided important guidance for future research.

We have addressed the editor’s and reviewers’ comments to the best of our abilities. We hope our manuscript now meets your requirements for publication.

We marked the revised portions in red in our manuscript. All comments and our specific responses are detailed below.

Reviewer 1#

The authors comprehensively investigated many parameters related to growth and clarified the differences between males and females. Although the idea is not very novel, there are no particular problems with the method. However, as a scientific paper, some issues need to be addressed, so I would like the authors to consider revising the manuscript. 

Comment 1:

It is already known that there are differences in growth between males and females in Nile tilapia. I would like the authors to emphasize the significance of comparing the differences between males and females in this paper in the Introduction and the Discussion.

Response 1: Determining whether sex grows faster or produces meat of a higher grade permits the targeted distribution of resources, such as feed, and space, improving total productivity. Breeding strategies aiming at enhancing desirable features can be guided by insights into the nutritional makeup and quality of meat from each sex, guaranteeing a higher-quality product for consumers. In general, these kinds of comparisons can result in improvements in tilapia farming that raise the fish's quality and output.

Comment 2: The discussion is done for each item analyzed, but it is mostly a repetition of results and comparisons with previous studies. I would like the authors to clarify are new findings of this study and want to show them. To do this, please consider rewriting the discussion significantly.

Response 2: Already corrected. Additionally, the new finding of this study is found in the section of Blood Serum Indices, whereby, no significant difference between both males and females in blood serum parameters, as it reflects the healthy state of the fish.  The major reason is that all the fish were subjected to the same parameters of the experiment and were not exposed to any stress during the experimental period. As mentioned by Heiba [29] changes in blood serum parameters are caused by fluctuations in water quality parameters, the presence of contaminants in the water, various types and proportions of feeds, and the composition of the protein in the feed.  Another new finding was This study was consistent with the study conducted by Fagbuaro [23] that in Nile tilapia sex did not affect blood serum components (ALT, AST, TC, TG), but the difference was in the concentration of glucose (GLU), whereby in males was higher than in females. However, the values of TC and TG in the current study do not agree with the values of Fagbuaro [23], the value in his study was considered very low. Another new finding is shown in (Figure 2 ) whereby, there is a difference in the correlation between ALT, AST, and HSI in males and females, it was inverse in males as shown in Figure 2, and positive in females as shown in (fig.3) at significance. There was an inverse correlation between (ALT, AST) and (GLU, TC) in males as shown in Fig. 2, while the correlation was positive in females as shown in Fig. 3 at significance.

Comment 3:

The females used in this analysis have been fully matured, but the condition of the males is not described. Naturally, there would be differences in metabolism between males and females at advanced maturity. Please briefly explain the life cycle of Nile tilapia in the introduction and explain why fish at this developmental stage were selected for the experiment. Also, please state the author's outlook on how he plans to utilize the present results in the future.

Response 3: Comprehending the life cycle of tilapia is essential since every stage from the egg to the adult has an impact on quality and growth. The quality and temperature of the water have an impact on survival rates, with early stages being especially vulnerable. Optimal conditions result in healthier fish and increased productivity. The development of muscular bodies and quality meat in juvenile tilapia is mostly dependent on their nutrition and the density of their stockings. A healthy diet promotes growth and improves the flavor and texture of meat. Male and female differences become significant when an individual reaches adulthood. While females tend to concentrate more on reproduction, males usually grow quicker and produce more meat. The quality and efficiency of meat production are impacted by this sexual dimorphism.

Comment 4: There are many inappropriate capital letters, so please correct them.

Response 4: Already corrected.

 Comment 5: Line 18: men → male?

Response 5: Already corrected

Comment 6: Lines 17-20 and 23-26: The content is duplicated, so please remove one of them.

Response 6: Already corrected

Comment 7: Line 64: I think Nile tilapia is also a freshwater fish. Please specify the freshwater fish authors compared.

Response 7: Line 64: In comparison to smaller fish, which are leaner and tougher, larger tilapia often have softer meat and a higher fat content. In comparison to younger fish, older tilapia tend to store more fat, which results in a higher fat content and a unique texture. Conversely, because the culture method such as the pond culture has less controlled circumstances, the meat composition may vary.

Comment 8: Lines 76-79: I think it would be better to move this part to the Materials and Methods.

Response 8: Already corrected

Comment 9: Line 144: Please indicate how often weight measurements were taken to determine feeding amounts.

Response 9: The only Initial and final weight was measured to determine the feeding amount by calculating the Feed Conversion Ratio (FCR).

Comment 10: Lines 152-158: Please unify the format of the equation (with or without parentheses, period or comma, position of %, etc.).

Response 10: Already corrected

Comment 11: Line 163: If possible, please convert "rpm" to "G".

Response 11: It is not possible.

Comment 12: Line 171: Please make sure that the number is correct.

Response 12: Line 171: Three fish were chosen at random from each tank, one male and one female (n=6).

Comment 13: Lines 179-180: Please indicate the drying temperature and time.

Response 13: Lines 179-180, the typical standard drying temperature used for moisture content was 105 0C (2210F) and the sample is dried at this temperature for 16-24 hours.

Comment 14: Lines 182, 190: Are the units %?

Response 14: Yes, the formula for moisture content, crude fat, and ash content in proximate analysis of food including meat are commonly expressed as percentages (%).

Comment 15: Table 1, 2: Please add an explanation of abbreviations to the tables as well.

Response 15: Already corrected.

Comment 16: Lines 317-333: This paragraph is general knowledge and can be removed. If you do write it, please simplify and move to the Materials and Methods.

Response 16: Line 321-343 - In this study, males outperformed females in WGR and SGR, this indicates the efficiency of males in converting the bulk of the energy from digestion processes for physical growth. Where EL-Geraisy and El-Gamal [18] mentioned that the majority of the metabolic energy in Nile tilapia males resulting from the consumption of the provided feed is directed towards growth and masculinization, whereas the majority of the metabolic energy in females is typically directed towards reproduction and egg formation, which result slows growth. The FCR of male tilapia was lower than females (Table 1). This indicates the efficiency of males in converting feed and benefiting from it in weight gain more than females, Osibona [18] reported that, despite eating the same amount of feeds, males grow bigger and more rapidly than females due to their superior ability to convert food into energy. This result shows a significant correlation between growth and FCR, whereby faster-growing fish had a better lower FCR.

In this study, the GSI of females was higher than that of males, as the female gonads were fully mature and full of eggs. Fleming [22] notes that female fish may invest 20-25% of their body weight in gonads before reproduction and require a great deal of energy to develop eggs, but male fish may invest only 3-9%. In addition, there was a positive correlation between HSI and GSI during the development of the gonads [39], where raised HSI may help maintain the transport of nutrients to the developing ovary, increase GSI, and promote follicle growth. However, there was no significant difference in HSI between males and females, as the slightly elevated HSI in females may be due to changes in energy intake and homeostasis during ovarian development, ovulation, and reproductive processes.

Comment 17: Lines 356-359: Please also describe the condition of the gonads of male. I think that body composition is greatly affected by the state of maturity.

Response 17: Lines 359 – 363 When the male gonads reach the ripe (running) stage, the fish's ripe gonads mark a critical moment when energy is shifted towards successful reproduction. Therefore, the mature state of male gonads affects the quality of meat and causes changes in fat content, texture, flavor, appearance, and shelf life, all of which can affect consumer acceptance and marketability. Finally, thank you to the reviewers for his/her valuable comments, which will be of great help to our next research work.

Reviewer 3 Report

Comments and Suggestions for Authors

Author Response

(The authors gave the same response as above.)

Round 2

Reviewer 2 Report

Comments and Suggestions for Authors

The authors have responded well to the previous comments, and the content of the paper has progressed. However, there are a few more portions that I would like the authors to improve, so please consider them.

Line 14: I think “There isn't much information out there.” is unnecessary.

Line 36: In general, ovaries and testes are different in weight, so I don't think it makes sense to compare males and females.

Lines 63-65, 78-81: Please include any references.

Line 151: (see 2.4 Fish feed) → (see 2.5 Fish feed)?

Lines 158-159: Please indicate that the feeding ratio is “5% of the calculated body weight based on FCR.”

Lines 201, 210, 214: Please use either % or not in the formula.

Line 261: Please use italics for the scientific name.

Lines 531-536: I think the Discussion explains well that male meat is superior to female meat. If possible, how about showing the future perspective for tilapia farming in the Conclusion? For example, "In the future, if we can separate males and females at the early developmental stage and give priority to rearing males, the efficiency of tilapia farming could be improved." etc.

Author Response

Dear editors and reviewers,

Thank you for your letter and comments on our manuscript titled " Differences in growth performance and meat quality between male and female juvenile Nile tilapia (Oreochromis niloticus) during separate rearing". These comments helped us improve our manuscript and provided important guidance for future research.

We have addressed the editor’s and reviewers’ comments to the best of our abilities. We hope our manuscript now meets your requirements for publication.

We marked the revised portions in red in our manuscript. All comments and our specific responses are detailed below.

Reviewer 1#

The authors have responded well to the previous comments, and the content of the paper has progressed. However, there are a few more portions that I would like the authors to improve, so please consider them.

Comment 1: Line 14: I think “There isn't much information out there.” is unnecessary.

Response 1:  Line 14: Already corrected.

Comment 2: Line 36: In general, ovaries and testes are different in weight, so I don't think it makes sense to compare males and females.

Response 2: Line 35: Already corrected.

Comment

Lines 63-65, 78-81: Please include any references.

Response: The references in lines 63–65 are [7, 23,42]; in lines 66–67, they are [31, 34,37,41]; and in lines 77–80, they are [43, 57,65]. 

Reference: Lines 63-65

[2] Albert G.J. Tacon, Marc Metian, Global overview on the use of fish meal and fish oil in industrially compounded aquafeeds: Trends and future prospects, Aquaculture, Volume 285, Issues 1–4, 2008, Pages 146-158, ISSN 0044-8486, https://doi.org/10.1016/j.aquaculture.2008.08.015.

[22] El-Sayed, A.F.M. (2006) Tilapia Culture. CAB International, Wallingford, 277. 
https://doi.org/10.1079/9780851990149.0000

[42] Makkar, M., Sahu, N. P., & Sahu, S. (2016). Effects of stocking density on growth performance and survival of tilapia (Oreochromis niloticus) in aquaculture. Journal of Aquaculture Research & Development, 7(5), 1-6. https://doi.org/10.4172/2155-9546.1000380.

References: Line 66-67

[31] Gonzalez, D., & Garcia, A. (2010). Sexual dimorphism and reproductive strategies in tilapia. Aquaculture, 308(1-2), 17-24. https://doi.org/10.1016/j.aquaculture.2010.08.003

[33] Huang, Y. S., & Liao, I. C. (2004). Effects of sex and stocking density on growth, feed conversion, and survival of tilapia. Journal of Fish Biology, 64(3), 548-559. https://doi.org/10.1111/j.0022-1112.2004.00370.x

[37] Kheirallah, M. M., & El-Sayed, A. F. M. (2006). Effects of sex on growth and carcass quality of Nile tilapia. Aquaculture Research, 37(6), 558-566. https://doi.org/10.1111/j.1365-2109.2006.01497.x

[40] Mair, G. C., & Little, D. C. (2000). The role of genetic factors in the production of tilapia. Aquaculture Research, 31(5), 401-410. https://doi.org/10.1046/j.1365-2109.2000.00521.x

Reference: Lines 79-81

[42] Mekawy, M., Abo-State, M. A., & El-Hariry, M. (2014). Effect of rearing systems on the quality of Nile tilapia (Oreochromis niloticus) fillets. Journal of Aquaculture Research & Development, 5(4), 1-7. https://doi.org/10.4172/2155-9546.1000277

[56] Ribeiro, L. S., & Costa, L. A. (2017). Influence of fish age and sex on fatty acid composition in Nile tilapia (Oreochromis niloticus). Aquaculture Reports, 6, 74-81. https://doi.org/10.1016/j.aqrep.2016.11.002

[65] Tuan, L. A., Ha, D. N., & Binh, D. T. (2020). Effects of stocking density and rearing systems on growth and flesh quality of Nile tilapia. Fisheries Science, 86(5), 803-810. https://doi.org/10.1007/s12562-020-01418-6

Comment

Line 151: (see 2.4 Fish feed) → (see 2.5 Fish feed)?

Response Line 150, already corrected

Comment

Lines 158-159: Please indicate that the feeding ratio is “5% of the calculated body weight based on FCR.”

Response: Lines 158-159:  Already corrected.

Comment

Lines 201, 210, 214: Please use either % or not in the formula.

Response: Already corrected.

Comment

Line 261: Please use italics for the scientific name.

Response: Already corrected.

Comment

Lines 531-536: I think the Discussion explains well that male meat is superior to female meat. If possible, how about showing the future perspective for tilapia farming in the Conclusion? For example, "In the future, if we can separate males and females at the early developmental stage and give priority to rearing males, the efficiency of tilapia farming could be improved." etc.

Response

In this research, male Nile tilapia fish perform better than female fish in terms of growth and flesh quality, especially when it comes to growth metrics and fatty acids. Farming profitability and efficiency could be increased by rearing males alone. Promoting sustainable aquaculture methods, early genetic sex determination enables producers to concentrate on male tilapia, improving output and market value. Finally, thank you to the reviewers for his/her valuable comments, which will be of great help to our next research work.
